# Dysregulated Interorganellar Crosstalk of Mitochondria in the Pathogenesis of Parkinson’s Disease

**DOI:** 10.3390/cells9010233

**Published:** 2020-01-17

**Authors:** Lara Sironi, Lisa Michelle Restelli, Markus Tolnay, Albert Neutzner, Stephan Frank

**Affiliations:** 1Division of Neuropathology, Institute of Medical Genetics and Pathology, University Hospital Basel, 4031 Basel, Switzerland; lisa.restelli@usb.ch (L.M.R.); markus.tolnay@usb.ch (M.T.); 2Department of Biomedicine, University Hospital Basel, University of Basel, 4031 Basel, Switzerland; albert.neutzner@unibas.ch; 3Department of Ophthalmology University Hospital Basel, University of Basel, 4031 Basel, Switzerland

**Keywords:** α-syn, LRRK2, DJ-1, Parkin, PINK1, ATP13A2, VPS35, MAM, mitophagy, neurodegeneration

## Abstract

The pathogenesis of Parkinson’s disease (PD), the second most common neurodegenerative disorder, is complex and involves the impairment of crucial intracellular physiological processes. Importantly, in addition to abnormal α-synuclein aggregation, the dysfunction of various mitochondria-dependent processes has been prominently implicated in PD pathogenesis. Besides the long-known loss of the organelles’ bioenergetics function resulting in diminished ATP synthesis, more recent studies in the field have increasingly focused on compromised mitochondrial quality control as well as impaired biochemical processes specifically localized to ER–mitochondria interfaces (such as lipid biosynthesis and calcium homeostasis). In this review, we will discuss how dysregulated mitochondrial crosstalk with other organelles contributes to PD pathogenesis.

## 1. Introduction

Parkinson’s disease (PD), the most common movement neurodegenerative disorder, is a complex multifactorial disease with an incidence range between 5 up to >35 per 100,000 population [1]. Clinically, it is characterized by motor symptoms such as bradykinesia, resting tremor, muscle rigidity, and postural instability, which may be accompanied by depression, sleep disorders, anosmia, and, with disease progression, dementia. The neuropathological hallmarks of the disease include a progressive loss of dopaminergic neurons in the substantia nigra (SN) pars compacta that project to the striatum, and the presence of α-synuclein (α-syn) positive neuronal inclusions known as Lewy bodies (LB) and Lewy neurites (LN) [2].

Familial and sporadic PD forms share common clinical, pathological, and biochemical characteristics. Although many aspects of PD pathogenesis remain elusive, dysregulation of various fundamental physiological processes has been implicated, including impairment of the ubiquitin-proteasome pathway, mitochondrial dysfunction, oxidative stress, and neuroinflammation.

Both environmental and genetic factors converge in the complex molecular pathophysiology of Parkinson’s disease, with mitochondrial dysfunction playing a major role [3,4,5,6]. A discussion of PD-associated risk factors is not the focus of our review. Several lines of evidence obtained from studies of familial forms of PD, patient tissue samples, and various in vitro/in vivo models point to a prominent involvement of dysregulated mitochondrial crosstalk with other organelles in addition to impaired mitochondrial quality control pathways. Here, we start from genes that have been linked to familial forms of PD to discuss the role of interorganellar crosstalk involving mitochondria (Table 1).

We specifically focus on how dysregulated communication of mitochondria with endoplasmic reticulum and lysosomes as well as compromised quality control at the mitochondrial level contribute to PD pathogenesis. For each interorganellar contact, we first provide a brief overview on their physiological organization and functions, and then describe how PD-linked genes affect these functions (Figure 1).

## 2. Mitochondria-Associated Membranes (MAMs)

The close apposition between ER and mitochondria was first described as an interorganellar contact by Bernhard in 1956 [7] and later by Copeland and Dalton, who, by electron microscopy, demonstrated the tight spatial relationship between these organelles in 1959 [8]. After performing fractionation studies, Jean Vance termed the biochemically distinct domains of the ER that are in close proximity to mitochondria MAMs (mitochondria-associated membranes), and showed that these specialized membrane contact sites contain the enzymatic activities involved in lipid transfer between ER and mitochondria for the biosynthesis of serine-containing phospholipids [9]. In electron microscopy studies, mitochondria were found to be in proximity to both smooth and rough ER tubules, with an interorganellar distance varying between 10 and 80 nm [8,10,11,12]. Different conditions, such as ER stress [13], metabolic state [12], and apoptotic stimuli [10] can affect the number, length, and/or width, as well as the protein composition [14] of these microdomains.

Reflecting their biochemical functions in lipid metabolism, MAMs are enriched in proteins such as phosphatidyl ethanolamine methyltransferase 2 (PEMT2), phosphatidylserine synthase 1 and 2 (PSS1/2) [15,16], and fatty acid CoA ligase 4 (FACL4). The latter, involved in triacylglycerol synthesis, is considered one of the most reliable MAM marker proteins [17].

Lipid synthesis, in particular the synthesis of triacylglycerol, phosphatidylcholine (PC), and phosphatidylethanolamine (PE), requires enzymatic activities associated with both ER and mitochondria. Phosphatidylserine (PS) is synthesized from PA by PSS1 in MAMs and is converted to PE by PS decarboxylase in mitochondria. One of the enzymes implicated in the final steps of PC synthesis, PEMT2 [18], was found to be restricted to MAMs [16].

Another enzyme located at ER–mitochondrial contact sites is acyl-CoA/diacylglycerol acyltransferase 2 (DGAT2), which catalyzes triacylglycerol synthesis and promotes lipid droplet formation [19]. MAMs are also enriched in further lipid metabolism enzymes, such as acyl-CoA/cholesterol acyltransferase 1 (ACAT1/SOAT1), which catalyzes the production of cholesterol esters that are subsequently incorporated into lipid droplets.

Besides their role in lipid metabolism, MAMs are also critically involved in Ca^2+^ homeostasis [20,21,22], as reflected by the enrichment of the Ca^2+^ channel inositol-1,4,5-triphosphate (IP3) receptor (IP3R) at these contact sites [23,24]. IP3Rs: type 3 is strongly enriched at MAMs [25]. Thus, MAMs represent Ca^2+^ signaling hubs providing ER-to-mitochondria Ca^2+^ transfer to maintain cellular bioenergetics, mitochondrial dynamics and transport, and also to modulate cell death decisions [26,27,28].

The stimulation of Ca^2+^ release from the ER through IP3Rs forms microdomains with high Ca^2+^ concentrations, which are important for the Ca^2+^ uptake into the mitochondrial matrix [20,29]. Mitochondrial Ca^2+^ uptake involves its diffusion across voltage-dependent anion channels (VDACs) of the outer mitochondrial membrane (OMM) [30] and the subsequent uptake through the low-affinity mitochondrial calcium uniporter (MCU), juxtaposed at the inner mitochondrial membrane (IMM) [31,32]. Indeed, Ca^2+^ concentrations modulate the enzymatic activities of mitochondrial ATP synthase and of the dehydrogenases that provide reducing equivalents to the respiratory chain [26]; they also regulate protein folding capacity, as ER chaperones depend on Ca^2+^ [33]. Ca^2+^ homeostasis is facilitated by cytosolic Ca^2+^ re-uptake into the ER through the sarco/endoplasmic reticulum (SR/ER) Ca^2+^ ATPase pump (SERCA) [34]. While Ca^2+^ fluxes enhance upon increased energy demand [35], excessive Ca^2+^ transfer can initiate programmed cell death through mitochondrial Ca^2+^ overload and opening of the mitochondrial permeability transition pore, leading to pro-apoptotic mediator release from mitochondria with subsequent effector caspase activation [36].

IP3R interacts with the OMM protein voltage-dependent anion channel isoform 1 (VDAC1) through glucose-regulated protein 75 (GRP75), a member of the Hsp70 family of chaperones, forming an interorganellar tethering complex between ER and mitochondria [22]. However, loss of IP3R does not interfere with ER–mitochondria association, which argues against an indispensable role of this Ca^2+^ channel in ER–mitochondria tethering [9]. As discussed below, additional ER–mitochondrial tethers exist.

Close physical, bidirectional interactions between ER and the mitochondrial network also play an important role in mitochondrial fission. The mitochondrial adaptors syntaxin 17, Mff, MiD49, and MiD51 that are involved in the recruitment of the fission-promoting dynamin-related protein Drp1 localize to ER–mitochondria interfaces [37,38]. ER tubules wrap around mitochondria, mediating constriction of the organelles at sites where subsequent mitochondrial division will occur [39]. Recent reports indicate that ER-bound inverted formin 2 (IFN2) mediates actin polymerization to promote mitochondrial fission [40]. Moreover, ER–mitochondria contact sites are spatially linked to actively replicating mitochondrial nucleoids, thereby coordinating mitochondrial DNA (mtDNA) synthesis with mitochondrial division to enable proper distribution of nucleoids between daughter mitochondria [41].

In addition, MAMs are also involved in the regulation of mitochondrial retrograde and anterograde transport along microtubules. In this context, at resting cytosolic Ca^2+^ concentrations, mitochondria move at maximal velocity, while their motility is reduced at IP3R-dependent Ca^2+^ hotspot regions, so that mitochondria accumulate and enhance local Ca^2+^ buffering by Ca^2+^ uptake which represents an important feedback mechanism in Ca^2+^ signaling [27].

The protein composition of some tethering complexes at MAM level continues to be a matter of debate. While Mitofusin 2 (MFN2), localized both on the ER and OMM, has been implicated in regulating ER–mitochondria juxtaposition, the field is still divided on the question of whether it functions as tether [42,43,44] or tethering inhibitor [45,46,47,48] (Figure 2).

Various proteins localized at ER–mitochondria interface such as PACS-2 [49] and GRP75 [22] affect organelle proximity upon modulation of their expression. It still remains unclear how these proteins mediate the tethering between the two organelle membranes. A direct role in tethering has been highlighted for a complex formed by Vesicle-associated membrane protein-associated protein B (VAPB) enriched in MAMs, and the OMM protein tyrosine phosphatase-interacting protein 51 (PTPIP51) [50]. In various biochemical assays, VAPB and PTPIP51 were shown to interact, and modulation of their expression (by siRNA knockdown or overexpression) affects ER–mitochondrial Ca^2+^ exchange and modulates interorganellar contacts, as assayed by EM. Beyond its Ca^2+^ exchange function, this ER-mitochondria tethering complex was also proposed to play a role in autophagy regulation [51]. In addition, an interaction of the VAPB-PTPIP51 complex with two other proteins which localize at the ER–mitochondria interface, oxysterol-binding protein (OSBP)-related protein 5 (ORP5) and OSBP-related protein 8 (ORP8), was shown recently [52].

### 2.1. MAMs in Parkinson’s Disease

MAMs serve crucial functions in various signaling pathways and metabolic processes, including mitochondrial bioenergetics and dynamics, Ca^2+^ homeostasis, and autophagy. While many of these functions are compromised in neurodegenerative disorders including Parkinson’s disease (PD), it is still unclear whether MAM dysregulation is cause or consequence of the pathogenic processes leading to neurodegeneration. Nevertheless, it seems clear that MAM dysfunction can accelerate neuronal death. Both changes in the number of contacts between ER and mitochondria, and impairments of their functionality have been associated with PD [53,54,55].

Mutations in several PD-associated genes have been causally related to mitochondrial dysfunction. Even if our current knowledge about the role of PD-related proteins in ER–mitochondria crosstalk is still far from complete, the following sections highlight their roles in maintaining MAM structure and function.

#### 2.1.1. α-Synuclein

*SNCA* was the first gene to be associated with familial cases of PD [56]. It encodes α-syn, a 14 kDa protein highly expressed in nervous tissues. On the cellular level, α-syn was found at presynaptic terminals where it is required for rapid and efficient clathrin-mediated synaptic vesicle endocytosis [57,58], reflecting a role in synaptic transmission. Beside its presence in the cytosol, a fraction of α-syn has been identified in mitochondria [59], where it is required for normal respiratory chain complex activity [60,61]. α-syn can influence Ca^2+^ exchange and the physical interaction between ER and mitochondria, as reported by different groups, with still-debated downstream effects [53,62,63].

α-syn presence has important implications for mitochondrial integrity: expression of either the α-syn disease mutation A53T at low levels, or of wild-type α-syn at high levels, result in fragmented mitochondria [64]. The mitochondrial fragmentation caused by α-syn mutations was reported to be independent of DRP1, as the function and recruitment of the fission protein to mitochondria was unaffected. It has been hypothesized that the increased mitochondrial fragmentation could be due to increased OPA1 cleavage, via an unknown mechanism [62].

Importantly, a portion of α-syn seems to be localized at MAMs [62], consistent with previous observations that the protein preferentially binds to lipid rafts [65] and to membrane domains rich in acidic phospholipids [66].

Pathogenic mutations of α-syn affect its binding to lipid membranes [67], as exemplified by the pathogenic A30P mutation, which decreases the amount of α-syn present in MAMs [65]. Decreased amounts of MAM-localized α-syn are also observed upon expression of the disease-causing mutation A53T [62], although in this case the ability of the mutant protein to bind to lipid membranes did not seem to be compromised [65]. It is known that this particular mutation makes the protein more prone to aggregation [68]. The reported net effect of both mutations (A53T, A30P) was a reduced amount of α-syn within MAMs and a concomitant increase of the mutant protein in the pure mitochondrial fraction, potentially leading to a decrease in mitochondrial membrane potential (MMP) [69] (Figure 3a).

This reduced MMP could promote OPA1 cleavage and consequently mitochondrial fragmentation [70]. In addition, decreased localization of both α-syn mutants at MAMs also reduced ER–mitochondria apposition, leading to impaired interorganellar crosstalk with compromised lipid synthesis; in fact, the conversion of PS into PE, a well-recognized biochemical MAM activity, was decreased upon mutant α-syn expression [62].

Although α-syn was shown to be a major component of Lewy bodies more than 20 years ago [71], subsequent proteomic studies revealed that LB consist of more than 300 proteins, of which around 90 were confirmed by immunohistochemistry [72]. Transmission electron microscopy (TEM) studies revealed that LB are composed of filamentous structures immunoreactive for α-syn [73]. More recently, Shahmoradian and colleagues, using correlative light and electron microscopy (CLEM), demonstrated that the vast majority of LB and LN actually consist of a crowded environment of membrane fragments, dysmorphic mitochondria and vesicular structures resembling lysosomes and autophagosomes, combined with non-fibrillar α-syn [74]. It has been hypothesized that these observations could reflect cellular attempts to segregate damaged lipid-based elements into aggresome-like structures. Indeed, LB were previously found to be immunoreactive for several markers of aggresomes [75], which form in response to cytoplasmic accumulation of misfolded protein [76,77].

#### 2.1.2. Parkin and PINK1

Aggregated proteins and damaged organelles are removed from the cytoplasm by autophagic mechanisms [78]. Mitophagy is a selective form of autophagy that mediates the removal of damaged mitochondria, thereby contributing to mitochondrial turnover [79]. Activation of this process is essential to protect neurons from pro-apoptotic proteins released by damaged mitochondria, which would otherwise trigger programmed cell death pathways in the cytosol [80].

Intriguingly, two PD-associated proteins, PTEN-induced putative kinase 1 (PINK1), a mitochondrially localized kinase, and Parkin, a cytosolic E3 ubiquitin ligase, are the two key players of this mitophagic quality control system. Mutations in PINK1 and Parkin are linked to early-onset familial PD [81], and extensive research efforts during the last decade have uncovered important aspects of the underlying pathogenic processes, some of which may also be shared with sporadic (idiopathic) PD.

Under basal conditions, PINK1 is imported into mitochondria through the translocase of the outer membrane (TOM) complex and then through the translocase of the inner membrane complex (TIM) into the matrix, where it is cleaved by the matrix processing peptidase and the inner membrane protease presenilin-associated rhomboid-like protease (PARL) [82,83,84]. Thereafter the cleaved product is released into the cytoplasm to be degraded by the proteasome via the N-end rule pathway [85]. However, in response to mitochondrial damage (loss of MMP or accumulation of misfolded proteins), PINK1 accumulates on the OMM. In addition to autophosphorylation, PINK1 phosphorylates Parkin, increasing its E3 ligase activity [86,87], and also phosphorylates pre-existing ubiquitin molecules at the mitochondrial surface [88]. Parkin is then thought to bind to phosphorylated ubiquitin, resulting in partial activation and tethering of Parkin to the OMM. The actions of PINK1 and Parkin contribute to amplification of ubiquitin phosphorylation, leading to conjugation of ubiquitin to several substrates [89]. The ubiquitinated cargo is then bound to specific autophagy receptor proteins that connect it to autophagosomes [90] which are formed at MAMs. In support of this model, upon stimulation of mitophagy, endogenous PINK1 was also found to be localized at MAMs.

Relevant to the ER-mitochondria interface, Parkin was also shown to ubiquitinate MFN2, VDACs and Miro [81]. BECN1/Beclin1 is required for the accomplishment of the mitophagic process, and silencing of this protein activates pro-apoptotic pathways [91]. Finally, autophagosomes fuse with lysosomes to complete the mitophagic process [92].

Fibroblasts from patients carrying mutated PINK1 or Parkin display increased ER–mitochondria juxtaposition, resulting in aberrant ER-to-mitochondria Ca^2+^ signaling [93,94]. Similar alterations were observed in mouse embryonic fibroblasts from Parkin knock-out mice and attributed to MFN2, which as a Parkin substrate is increased at the MAM fraction upon Parkin dysfunction [94] (Figure 3c).

Parkin and PINK1 null mice generally fail to recapitulate the degeneration of dopaminergic neurons in the SN [95,96,97]. Furthermore, loss of Parkin does not worsen the neurodegenerative phenotype of MitoPark mice [98]. Nevertheless, Parkin activity is critical for the survival of nigral dopaminergic neurons in Mutator mice (homozygous for a proofreading deficiency in DNA polymerase γ) which have accelerated mtDNA mutation rates [99].

Parkin was reported to co-regulate ER–mitochondria contact sites together with the transcription factor peroxisome proliferator-activated receptor γ coactivator 1α (PGC-1α), a key modulator of mitochondrial biogenesis [100]. Loss of Parkin function results in the accumulation of the zinc finger transcriptional repressor Parkin interacting substrate (PARIS), which suppresses PGC-1α-dependent transcription. Postmortem analysis of SN tissue of PD patients validated this finding, with dopaminergic neurons displaying reduced PGC-1α levels [101].

Parkin has a Ser65 residue within its N-terminal ubiquitin-like (UBL) domain, similar to that of ubiquitin. This residue is phosphorylated by PINK1, resulting in an open and active conformation [102,103]. Characterization of primary cells derived from two unrelated, early-onset PD patients with homozygous Parkin Ser65Asn (ParkinS65N) mutation demonstrated that this mutant is inactive, suggesting that the loss of PINK1-dependent Parkin Ser65 phosphorylation and subsequent inactivation in humans is sufficient to cause PD [104].

PINK1-deficiency in Drosophila, mouse models and patient-derived cells resulted in mitochondrial complex I defects [105] and decreased mitochondrial membrane potential [106], associated with loss of Ser250 phosphorylation of the complex I subunit NADH ubiquinone oxidoreductase subunit A10 (NDUFA10).

#### 2.1.3. DJ-1

The DJ-1 protein serves a broad variety of functions. It plays an essential role in sensing and reacting to oxidative stress, thereby protecting cells against reactive oxygen species (ROS) [107,108]. Within its active site, DJ-1 contains an essential cysteine residue that functions as an oxidative stress sensor. Beyond this function, DJ-1 neutralizes ROS [107,109]: mitochondria-localized DJ-1 is a component of the thioredoxin/apoptosis signal-regulating kinase 1 (Trx/Ask1) complex, which regulates the clearance of endogenous ROS through activation of the radical scavenging system [110]. Brains of PD patients contain high levels of oxidized DJ-1, which indicates an increased ROS scavenging activity [111,112].

In addition to oxidative stress, DJ-1 protects against other toxic agents by modulating PTEN activity and Akt signaling [113,114], either by interacting with the MAPK kinase cascade [115], the p53 pathway [114,116], or by stabilizing the antiapoptotic Bcl-XL protein [117]. Crystallography revealed that DJ-1 is a homodimer, which appears to be critical for its physiological function [118,119,120].

DJ-1 protein localizes at MAMs where it modulates ER–mitochondria interactions and consequently Ca^2+^ transfer between the two organelles, thereby maintaining mitochondrial function and structure. Depletion or lack of function of this protein causes alterations of mitochondrial morphology, decreases mitochondrial membrane potential, reduces ER-to-mitochondria Ca^2+^ transfer and impairs mitochondrial motility in neurites [55].

Whereas mutations in the gene encoding DJ-1 (PARK7) lead to familial early-onset PD, the exact mechanisms underlying its role in PD pathogenesis still remain elusive [121,122]. In in vitro systems as well as in living cells, DJ-1 interacts directly with monomeric and oligomeric α-syn [123]. The same study showed that familial DJ-1 mutations (L166P, M26I, L10P and P158Δ) abrogated its interaction with α-syn, which could be due to the low expression levels of DJ-1 mutants, as they are more rapidly degraded than wild-type DJ-1 protein [120]. Furthermore, the above-mentioned DJ-1 mutants correlate with increased α-syn oligomerization, suggesting a loss of DJ-1 chaperone function [123]; (Figure 3b).

#### 2.1.4. LRRK2

Leucine-rich repeat kinase-2 (LRRK2) is a large, multi-domain protein involved in a number of functions, such as GTP hydrolysis, kinase activity, and protein binding. Even though its cellular function is largely unknown, emerging evidence attributes to LRRK2 roles in autophagic regulation, microtubule dynamics, and mitochondrial function. In addition to being localized mainly to the cytoplasm, some LRRK2 also resides at mitochondria [124].

Autosomal dominant LRRK2 mutations are associated with both familial and sporadic PD [125,126]. Expression of mutant LRRK2 induces several negative effects at the mitochondrial level, such as increased fragmented mitochondria that produce more ROS and less ATP, leading to increased cell vulnerability to stressors. Skin biopsies from patients carrying the G2019S mutation, which results in an increased kinase activity of the protein, show reduced mitochondrial membrane potential, aberrant organelle morphology, and decreased total intracellular ATP levels [127]. It still remains unclear how this increased kinase activity impairs cellular functions and promotes cell death [128].

LRRK2 interacts with a number of mitochondrial fission/fusion regulators, either in the cytosol or on mitochondrial membranes [129]. It was shown that LRRK2 associates with Drp1, the key mediator of mitochondrial fission. Neuronal expression of two LRRK2 mutants, G2019S and R1441C, led to increased interaction with DRP1 and higher phosphorylation levels of the fission protein, resulting in mitochondrial fragmentation and enhanced ROS levels [130,131] (Figure 3d).

LRRK2 also interacts and modulates the activities of the mitochondrial fusion regulators MFN1, MFN2, and OPA1. Indeed, PD patients carrying the G2019S mutation showed decreased levels of L-OPA1 [129]. Moreover, fibroblasts and neuroblastoma cells expressing the G2019S mutant display increased basal oxygen consumption and a decreased mitochondrial membrane potential, potentially due to a proton leak caused by upregulation of mitochondrial uncoupling proteins 2 and 4 (UCP2, UCP4) [132]. Thus, the effect of increased LRRK2 activity is decreased mitochondrial fusion with concomitantly increased fission of the organelles, suggesting that LRRK2 may be an important modulator of mitochondrial dynamics.

LRRK2 kinase activity also regulates ER–mitochondrial tethering by modulating the PERK-dependent ubiquitination pathway under ER stress conditions. In this context, LRRK2 interacts with the E3 ubiquitin ligases MARCH5, MULAN, and Parkin, thereby blocking PERK-mediated phosphorylation and activation of these E3 ubiquitin ligases. Kinase-active LRRK2 (G2019S) dissociates from ER ubiquitin ligases, allowing PERK to phosphorylate and thereby activate these enzymes towards MAM components, impinging on ER–mitochondrial tethering [133].

Another crucial aspect of mitochondrial dynamics in the context of neurodegeneration is mitochondrial trafficking, where mitochondrial locomotion is tightly controlled to preserve energy homeostasis. Before the initiation of the mitophagy cascade, mitochondrial motility halts, enabling the sequestration of damaged mitochondria. This arrest of depolarized mitochondria is achieved by removal of the Miro protein from the mitochondrial surface; this process is promoted by LRRK2, which forms a complex with Miro, targeting it for PINK1/Parkin-dependent degradation [134]. Interestingly, the LRRK2 mutant G2019S disrupts its interaction with Miro, slowing down Miro degradation and mitochondrial arrest, consequently delaying mitophagy [124].

## 3. Mitochondria–Lysosome Membrane Contact Sites

Lysosomes, together with mitochondria, are critical for the maintenance of cellular homeostasis, as reflected by the fact that dysfunction of both organelles is functionally and genetically linked to several human diseases [135,136,137,138]. Similar to mitochondria, lysosomes are highly dynamic organelles that are responsible for the turnover of cellular components, including proteins and lipids, via mature enzymes stored in the lysosomal lumen. In addition, these organelles also act as iron and calcium stores. Furthermore, they can mediate cell death signaling upon lysosomal membrane permeabilization [139].

Whereas numerous reports have demonstrated indirect functional interactions between mitochondria and lysosomes [140,141,142,143,144,145,146,147], studies focusing on lysosomal degradation of mitochondria either through mitophagy [81] or via fusion of mitochondrial-derived vesicles (MDVs) with lysosomes [148] showed a direct interaction between these organelles upon cellular stress [149]. The mitophagic process can occur via mitophagy receptors (Optineurin and NDP52) which are recruited in a PINK1/Parkin-dependent manner to ubiquitinated mitochondria, which are then targeted through LC3 to the autophagosome [150,151]. Alternatively, MDVs [148] are small vesicles that bud off from mitochondria and contain distinct subsets of OMM and mitochondrial matrix proteins. MDVs generated in a PINK1/Parkin-dependent manner are targeted to lysosomes, to selectively degrade a subset of mitochondrial proteins instead of entire mitochondria [152].

Mitochondria–lysosome contact sites have been imaged in various cell types under healthy conditions taking advantage of different techniques, such as 2D and 3D electron microscopy [153,154], correlative light electron microscopy (CLEM) [153], CLEM combined with focused ion beam scanning electron microscopy (FIB-SEM) [155], lattice light sheet spectral imaging [156], and structured illumination microscopy [153,157,158]. The average distance between mitochondrial and lysosomal membranes is ~10 nm [153,154], and approximately 15% of lysosomes are in contact with mitochondria at any time point, with contact sites remaining stably tethered for an average of 60 s [154,157]. These contact sites do not represent autophagosome biogenesis events or mitophagy, given their negative staining for autophagosome markers [153]. Furthermore, knockout of five autophagy receptors (NDP52, OPTN, NBR1, TAX1BP1, and p62) did not prevent mitochondria–lysosome contact formation [158]. Moreover, mitochondria involved in these contacts were distinct from MDVs as they contained intermembrane space and mitochondrial matrix proteins, and were larger (over 500 nm) compared to MDVs (about 100 nm) [148,153].

The small GTPase Rab7 is a master regulator of lysosomal maturation, positioning, and network dynamics [159]. As evidence of the importance of lysosomal dynamics, mutations in Rab7 lead to peripheral neuropathy in humans [160,161,162,163]. Rab7 modulates mitochondrial–lysosome tethering and untethering through its ability to alternate between an active, lysosomal-localized GTP-binding state, and an inactive, cytosolic GDP-binding state. Lysosomal GTP-bound Rab7 promotes tethering via lysosomal membrane-bound Rab7 effector proteins [153]. Then Rab7 GTP hydrolysis mediates the untethering, involving the recruitment of cytosolic TBC1D15 (Rab7 GAP) to mitochondria via the OMM protein Fis1 [164], where it can interact with lysosomal GTP-bound Rab7 to promote GTP hydrolysis. GDP-bound Rab7 is no longer able to bind Rab7 effectors and loses its localization to the lysosomal membrane [165] leading to untethering of the two organelles. Contact sites between mitochondria and lysosomes are also able to modulate mitochondrial dynamics, as most mitochondrial fission events (>80%) are marked by LAMP1-positive vesicles but not early endosomes or peroxisomes [153].

### 3.1. VPS35

Vacuolar sorting protein 35 (VPS35) is a key component of the retromer complex, involved in intracellular protein trafficking. VPS35 mediates retrograde delivery of cargo from endosomes to Golgi, as well as recycling endosomal cargo to the cell surface [166,167].

The retromer can be divided into a cargo-selective complex (CSC) trimer composed of VPS26, VPS29 and VPS35, involved in binding and sorting protein cargo [168,169], and a sorting nexin (SNX) dimer, consisting of SNX1 or SNX2 and SNX5 or SNX6 in mammalian cells (SNX5 and SNX17 in yeast). These SNX proteins are members of the SNX-BAR family and function in retromer association with the endosomal membrane through a Bin-Amphiphysin-Rvs (BAR) and phox homology (PX) domain [166,168].

VPS35 has a role in the formation of MDVs, which shuttle cargo from mitochondria to either peroxisomes or lysosomes, being so involved in mitochondria quality control [148,170]. This component of the retromer was found to interact with DRP1 and implicated in mitochondrial DRP1 complex recycling and mitochondrial fission. Indeed, DRP1 complexes are present on the OMM where they remain with daughter mitochondria after fission [171]. These complexes probably become inhibitory for subsequent fission events, owing to the occupancy of fission sites or to the sequestration of DRP1 recruiting factors [172,173,174,175,176,177]. Through the interaction between VPS35 and DRP1, the retromer mediates DRP1 complex removal from mitochondria to lysosomes or peroxisomes via the formation of MDVs, diminishing their inhibitory effects on mitochondrial fission [178].

Remarkably, the VPS35 D620N mutation is associated with autosomal-dominant PD [179,180]. PD patient fibroblasts expressing this mutated protein showed fragmented and functionally impaired mitochondria. These alterations were accompanied by an increased VPS35–DRP1 interaction leading to an enhanced turnover of mitochondrial DRP1 complexes through MDVs and lysosomal degradation [178].

VPS35 can also impinge on mitochondrial dynamics by an MFN2-dependent mechanism. A proteomic study suggested that VPS35/retromer interacts with the OMM E3 ubiquitin ligase MUL1 (also known as mitochondrial-anchored protein ligase, MAPL) [181]. VPS35 promotes the degradation of MUL1, which would otherwise degrade MFN2. Accordingly, the PD-linked VPS35 D620N mutation increases MUL1-mediated MFN2 degradation [182] (Figure 4a).

### 3.2. ATP13A2

The *PARK9* gene encodes the protein ATP13A2, a transmembrane lysosomal type 5 P-type ATPase [183], which has been linked to a neurodegenerative disorder known as Kufor–Rakeb syndrome (KRS), as well as to some juvenile and early-onset forms of PD [183,184,185,186,187]. Several studies focused on determining the cationic substrate of this transporter. While mammalian cell models supported Mn^2+^-modulating activity of ATP13A2 [188,189,190], studies using KRS patient-derived cells revealed Zn^2+^ dyshomeostasis [191,192,193] causing abnormal mitochondrial and lysosomal metabolism, with dysfunctional energy production and reduced lysosomal proteolysis, respectively.

The analysis of fibroblasts from two patients with the L3292 and L6025 ATP13A2 mutations showed an impaired clearance of autophagic vacuoles, accompanied by impaired lysosomal acidification, cathepsin activity, and proteolytic capacity, while the delivery of substrates to lysosomes by either macroautophagy or chaperone-mediated autophagy (CMA) translocation did not seem to be affected [194]. α-syn can be degraded by lysosomal pathways, such as macroautophagy and CMA, as well as by the proteasome [195,196].

PD-linked mutations in ATP13A2 may result in insufficient clearance of α-syn through lysosomes, resulting in its accumulation in the cytosol. Furthermore, postmortem nigral tissue samples from sporadic PD patients exhibited decreased neuronal levels of ATP13A2, which appeared to be mostly trapped in Lewy bodies [194] (Figure 4b).

### 3.3. LRRK2

Beside its involvement at MAMs (described above), LRRK2 serves a critical role in the autophagic pathway at the lysosomal level. During autophagy, damaged organelles and aggregated proteins are engulfed within autophagosomes, subsequently delivered to the lysosome for degradation [197,198]. Local lysosomal release of calcium is required for autophagosome–lysosome fusion [199]. Any disruption affecting autophagosome formation, fusion of autophagosomes with amphisomes or lysosomes, hydrolytic degradation, or the re-formation of lysosomes can impair the autophagic process, resulting in accumulation of autophagy substrates and structures [197,198].

Lysosomal function and protein degradation are regulated by many factors, such as lysosomal pH [200], calcium release [199], and membrane trafficking [201]. Lysosomal dysfunction was shown to lead to α-syn accumulation [202], which could play a role in the formation of Lewy bodies, the pathological hallmark of PD. Furthermore, LRRK2 has been implicated in lysosomal pH regulation [203,204], which is critical for the activity of degradative enzymes and for the fusion of autophagosomes and lysosomes [205]. The authors of [206] investigated the role of LRRK2 in lysosome biology and the autophagy pathway in primary neurons by expressing human wild-type LRRK2 (hWT-LRRK2) and the human LRRK2-G2019S or LRRK2-R1441C mutations, and demonstrating that mutations in different enzymatic domains elicit different effects on LRRK2 enzymatic activity. Neurons expressing hWT-LRRK2 or LRRK2-G2019S displayed a decreased rate of autophagosome formation, which was dependent on LRRK2 kinase activity.

In contrast, neurons expressing LRRK2-R1441C displayed a significantly increased lysosomal pH and alterations in lysosomal calcium dynamics, resulting in impaired autophagosome–lysosome fusion and decreased lysosome-mediated degradation (Figure 4c). These latter effects occurred independently of LRRK2 kinase activity. It is interesting to note here that hWT-LRRK2 interacts with the a1 subunit of the v-type H+ ATPase proton pump (vATPase a1), responsible for the regulation of lysosomal pH. Conversely, LRRK2-R1441C loses this interaction, leading to dysregulated vATPase a1 protein expression and cellular localization, and resulting in impaired autolysosome maturation [206].

## 4. Perspective

Over decades, research on PD pathogenesis has been dominated by a focus on mitochondrial bioenergetic defects, oxidative stress, and cell death mechanisms. With the discovery in 1997 that Lewy bodies are composed of misfolded/aggregated α-syn [72] and that mutations in the α-syn gene were linked to some inherited forms of the disease [57], the attention of the field has increasingly shifted towards the mechanisms of abnormal protein aggregation and spreading of α-syn pathology. The last years have further improved our understanding of the disease; in particular, the pathogenic importance of properly regulated interorganellar crosstalk was increasingly recognized. Relatively recent insights into dysregulated crosstalk of mitochondria with the endoplasmic reticulum and lysosomes may provide the foundation for a more unifying picture that could help to explain how mitochondrial dysfunction, bioenergetic defects, abnormal protein aggregation, and neuronal cell death converge in PD pathogenesis.

Clearly, our understanding of the complex molecular mechanisms underlying PD pathogenesis and progression is still far from complete, and crucial questions remain to be answered. Among these, it remains to be clarified what event(s) initiate(s) PD pathogenesis, how Lewy bodies form, which of the intracellular functions of α-syn are actually relevant for disease onset and progression, and what role the microbiome plays in modulating PD, to name but a few.

As research efforts in this field increasingly focus on interorganellar communication as opposed to single organelle biology [207], we expect that the picture of PD pathogenesis will become more defined in the near future.

## Figures and Tables

**Figure 1 cells-09-00233-f001:**
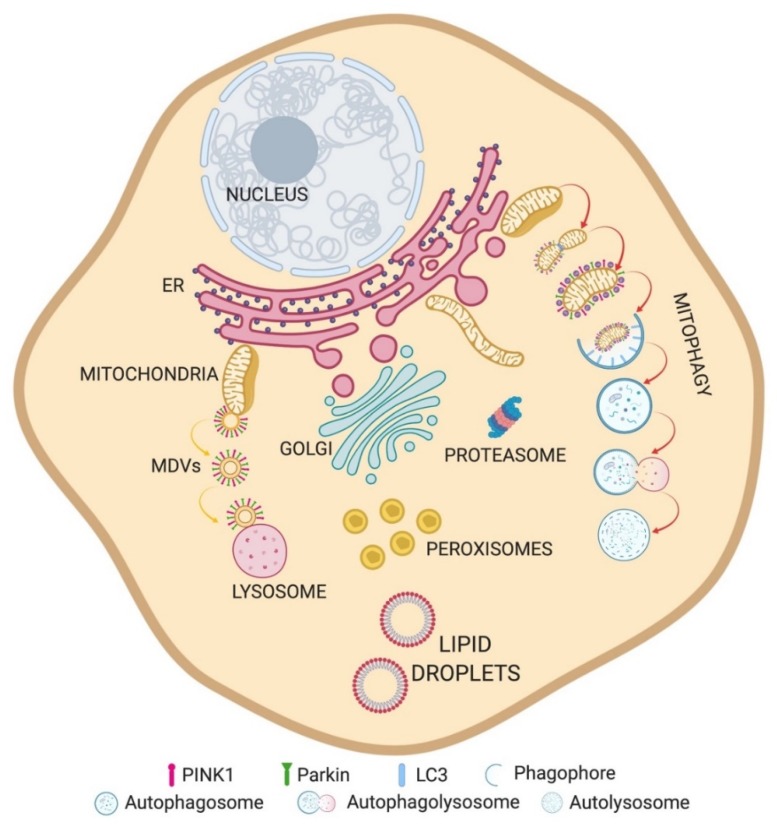
Overview of organelle crosstalks. Schematic representation of organelles and their relationships.

**Figure 2 cells-09-00233-f002:**
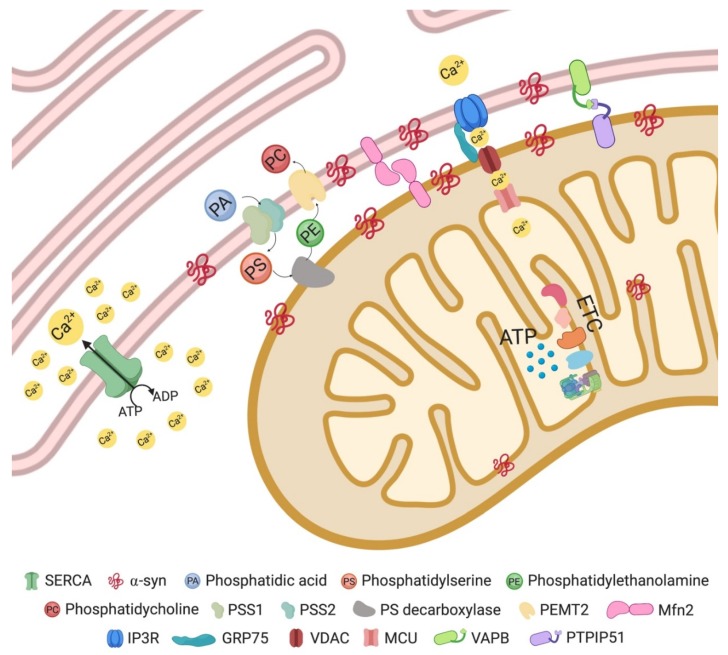
Mitochondria–ER contact site and main resident proteins (see text for details).

**Figure 3 cells-09-00233-f003:**
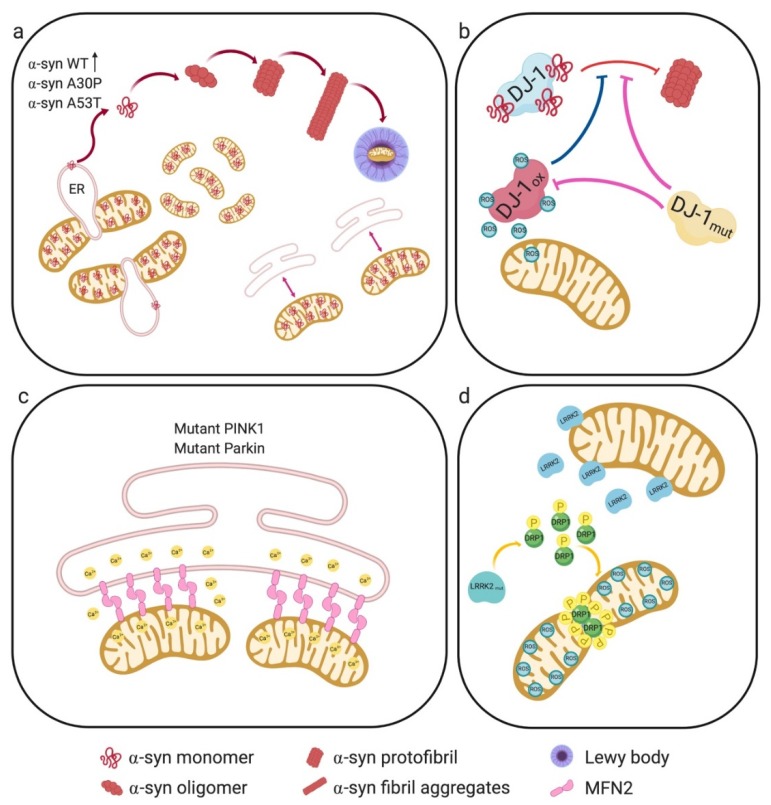
PD-associated genes and their roles in Mitochondria-associated membrane (MAM) structure and function. (**a**) Mutant α-syn results in DRP1-independent mitochondrial fragmentation, reduced MAM-associated mutant α-syn, with a concomitant increase in the pure mitochondrial fraction. This results in reduced ER–mitochondria apposition, leading to impaired interorganellar crosstalk. The A53T mutation makes the protein more prone to aggregation. (**b**) DJ-1 responds to oxidative stress, protecting cells against ROS. DJ-1 interacts with monomeric and oligomeric α-syn, preventing its oligomerization. Under oxidative stress conditions, oxidized DJ-1 is unable to interact with α-syn and to prevent its oligomerization. Likewise, DJ-1 mutations also abrogate its interaction with α-syn and no longer neutralize ROS. (**c**) Mutant PINK1 or Parkin increase ER–mitochondria juxtaposition, resulting in aberrant ER-to-mitochondria Ca^2+^ signaling. Furthermore, Parkin dysfunction could lead to increased levels of its substrate MFN2 at MAMs. (**d**) LRRK2 mutations increase its interaction with DRP1, and enhance DRP1 phosphorylation. This results in mitochondrial fragmentation, enhanced ROS, and decreased ATP levels.

**Figure 4 cells-09-00233-f004:**
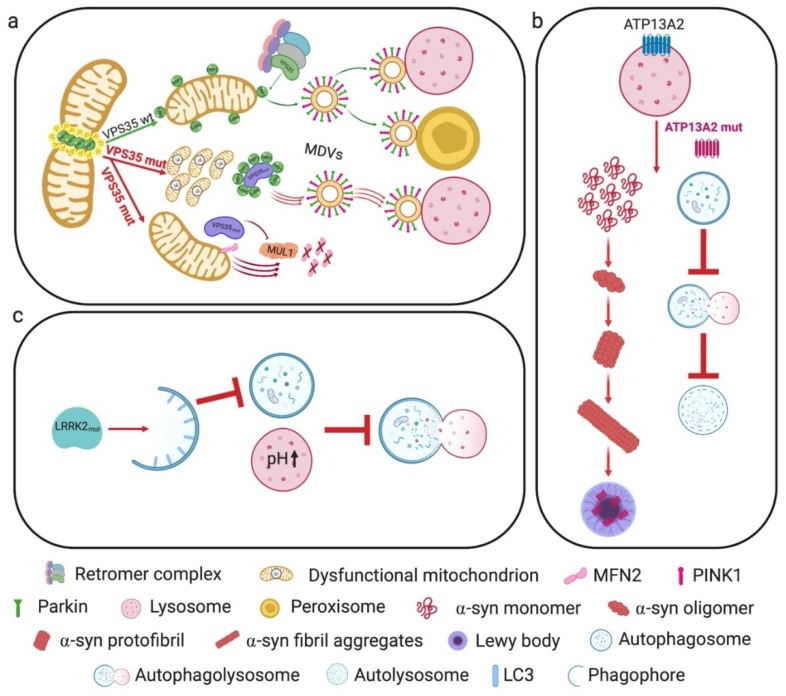
PD-associated genes and their roles in mitochondria-lysosome crosstalk. (**a**) VPS35 is a key component of the retromer complex involved in the removal of DRP1 complexes from mitochondria to lysosomes or peroxisomes through MDVs. Mutant VPS35 enhances turnover of mitochondrial DRP1 complexes through MDVs and lysosomal degradation, accompanied by fragmented and dysfunctional mitochondria. PD-linked VPS35 mutant also leads to increased MUL1-mediated MFN2 degradation. (**b**) ATP13A2 mutations impair the autophagic process, leading to the cytosolic accumulation of α-syn. Sporadic PD patients show decreased levels of this protein, which is also found in Lewy bodies. (**c**) Mutant LRRK2 protein impinges on autophagosome formation, alters lysosomal pH and lysosomal calcium dynamics, resulting in impaired autophagosome–lysosome fusion and lysosome-mediated degradation.

**Table 1 cells-09-00233-t001:** Parkinson’s disease (PD)-linked genes related to mitochondria interorganellar contacts.

HGNC ID	Gene Symbol	Alternative Designation	Chromosomal Location
Mitochondria-associated membranes (MAMs)			
HGNC:11138	SNCA	α-synuclein	4q22.1
HGNC:8607	PRKN	Parkin	6q26
HGNC:14581	PINK1	PTEN-induced putative kinase 1 (PINK1)	1p36.12
HGNC:16369	PARK7	DJ-1	1p36.23
HGNC:18618	LRRK2	Leucine-rich repeat kinase-2 (LRRK2)	12q12
Mitochondria-lysosome contact sites			
HGNC:13487	VPS35	Vacuolar sorting protein 35 (VPS35)	16q11.2
HGNC:30213	ATP13A2	ATPase 13A2	1p36.13
HGNC:18618	LRRK2	Leucine-rich repeat kinase-2 (LRRK2)	12q12

Overview of PD-linked genes role in interorganellar crosstalk involving mitochondria. HGCN IDs are in accordance with the *HUGO Gene Nomenclature Committee at the European Bioinformatics Institute* (HGNC) (https://www.genenames.org).

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
