# Peer review of "Dysregulated Interorganellar Crosstalk of Mitochondria in the Pathogenesis of Parkinson’s Disease"

_cells, 2020, doi:10.3390/cells9010233_

Round 1

Reviewer 1 Report

Sironi et al. have summarized what is known about changes in interorganellar communication in PD. The paper is very well written and contains several high-quality figures. To better convey the importance of the mechanisms described in the manuscript, it would be advisable to put the observation into a bigger context. Some additional minor suggestions are listed below. Overall, this is a great addition to the PD literature that will be cited extensively.

Specific Points:

What are the determinants of PD incidence other than genetics. A quick reference might help the reader understand if risk factors outside the genetic realm might impact organellar contact sites. Overall, it is not clear yet how important changes in mitochondria are for the disease. Are they always observed? Are they always causative? What other mechanisms could play a role? Are alternative causes for PD (e.g., a-synuclein accumulation or inflammation) also impacting mitochondria or membrane contact sites? In summary, are the described causative mechanisms central or peripheral to the disease? The review does not mention the discovery of ER-mitochondria contacts by Wilhelm Bernhard, the first reference of them in 1952, much earlier than Copeland and Dalton. When mentioning IP3Rs, which ones do the authors mean? There are three of them. How could calcium mechanistically control tethering? Some Rabs implicated in PD are not described. In particular, Rab7L1, Rab38 and Rab32 are known to give rise to contact site changes, and are also known to be mutated in PD. These should be discussed. What is the relative importance of changes occurring upon a-synuclein mutations? What is known about the relative changes on different mechanisms for individual mutations? (Have these been looked at comprehensively in the respective studies?) Are there changes in mitochondria-lysosome calcium crosstalk that could impact on PD?

Author Response

We thank the referee for his comments, some of which we feel are rather suggestions and points to consider for future work. In particular, we intentionally refrained from discussing determinants (i.e. risk factors) other than genetics as most of them, such as age, are not PD-specific. For others such as exposure to certain toxins, a large body of literature already exists, and a discussion of those aspects would simply be out of scope of our review. Likewise, mutations in GBA are well-known to increase the risk of PD. Irrespective of whether age, toxins, GBA mutations or other predisposing factors: their effects on specific membrane contact sites have not (yet) been studied systematically (the field of membrane contact site research is relatively new). We therefore now briefly mention in the manuscript that a discussion of PD-associated risk factors is not the focus of our review (page 1)

We have now mentioned the Bernhard reference of 1956, as suggested by the reviewer (page 3 of revised manuscript, beginning of section 2).

With Rab proteins the situation is similar to what we alluded to above: most reports on their potential functions are rather circumstantial, and while some of them such as Rab32, Rab38, and Rab7L1 have been implicated in PD pathogenesis, their functions at specific contact sites have not (yet) been studied systematically or are less clear in the context of PD  (e.g. Rab7L1 has even been described as decreasing PD risk (Gan-Or Z. et al 2012, PMID 22232350; Guo XY at al, 2014, PMID 25040112; Goudarzian M. et al, 2015, PMID 26344175).

As for IP3Rs: type 3 is strongly enriched at MAMs compared to the other 2 types (Morciano et al, 2018 PMID 29626751);as suggested by the referee, this is now mentioned in the manuscript (page 4).

Most of the papers concerning this protein at MAMs refer to it as IP3R or IP3Rs.

The relative importance of changes occurring upon a-synuclein mutations is described at page 6 of the manuscript (section 2.1.1 last paragraph) for the two mutations affecting MAMs. To the best of our knowledge, the relative changes on different mechanisms for individual mutations have not been looked at comprehensively.Lastly, mitochondria-lysosome crosstalk is a relatively new and emerging field, and the mechanisms governing the functional crosstalk between the two organelles remain to be elucidated.

Reviewer 2 Report

In this manuscript, Sironi et al. summarize recent knowledge on the role of dysregulated mitochondrial crosstalk with other organelles in Parkinson’s disease (PD) pathogenesis. Although a similar topic has been published (e.g. Gomez-Suaga et al., Cell Death Dis, 2018, PMID 29497039; Ploteqher and Duchen, Front Cell Dev Biol, 2017, PMID 29312935), this manuscript focuses on PD-associated gene products and updates very recent data. Overall, the manuscript is well studied and organized. There are a few comments to help to improve the quality of the manuscript.

Section 2.1.3: This section does not describe the relationship between DJ-1 and MAMs in PD pathogenesis. The following papers report the role of DJ-1 in MAM structure/function: Parrado-Fernandez et al., 2018, J Cell Mol Med, PMID 30133157; Ottolini et al., 2013, Hum Mol Genet, PMID 23418303. Line 290: “Section 2.1.3” should be “2.1.4”. Also, Toyohuku et al. have recently reported the role of LRRK2 in ER-mitochondria tethering (Toyohuku et al., 2019, EMBO J, PMID 31821596). This paper provides a new insight into MAM-associated PD pathogenesis. Sections 3.2 and 3.3: These sections lack description of the roles of ATP13A2 and LRRK2 in mitochondria-lysosome crosstalk, and of the molecular mechanism in which dysregulated mitochondria-lysosome crosstalk undergoes progression of PD.

Author Response

We thank the referee for his valuable comments.

We have now discussed the function of DJ-1 in MAM structure/function on page 9 of the revised Ms (3rd paragraph of section 2.1.3.). We have now also cited the work by Parrado-Fernandez (2018) and by Ottolini (2013), as suggested by the referee.

In addition, with regard to the referee`s comment on a new report on LRRK2`s role in ER-mitochondrial tethering, we have now added a respective paragraph (page 10, 5th paragraph of section 2.1.4) and cited the new work by Toyohuku (2019).

Finally, we have also elaborated in more detail on the role of ATP13A2 in mitochondrial-lysosome crosstalk (section 3.2., page 13).